# The Emborrhoid Technique for Treatment of Bleeding Hemorrhoids in Patients with High Surgical Risk [note 1]

**DOI:** 10.3390/jcm11195533

**Published:** 2022-09-21

**Authors:** Paola Campennì, Roberto Iezzi, Angelo Alessandro Marra, Alessandro Posa, Angelo Parello, Francesco Litta, Veronica De Simone, Carlo Ratto

**Affiliations:** 1Proctology Unit, Fondazione Policlinico Universitario Agostino Gemelli—IRCCS, 00168 Rome, Italy; 2Dipartimento di Diagnostica per Immagini, Radioterapia Oncologica ed Ematologia, Istituto di Radiologia, Fondazione Policlinico Universitario Agostino Gemelli—IRCCS, 00168 Rome, Italy; 3Department of Medicine and Translational Surgery, Università Cattolica del Sacro Cuore, 00168 Rome, Italy

**Keywords:** Emborrhoid technique, arterial embolization, hemorrhoidal disease, hemorrhoidal bleeding

## Abstract

The Emborrhoid is an innovative non-surgical technique for the treatment of severe hemorrhoidal bleeding. Patient selection and the impact on quality of life have not been fully investigated. This prospective observational study aims to evaluate the clinical outcomes after Emborrhoid in patients with high surgical risk. All patients with high surgical risk and anemia due to hemorrhoids were enrolled. Clinical data and previous blood transfusions were collected. The Hemorrhoidal Disease Symptom Score and Short Health Scala were completed before the procedure and during the follow-up visits at 1, 6 and 12 months. Transfusions and serum hemoglobin level variations were registered. Perioperative complications and the recurrence of bleeding were assessed. Trans-radial/femoral embolization of superior rectal artery, and/or middle rectal artery was performed with Interlock and Detachable Embolization Coils. From September 2020 to February 2022, 21 patients underwent a superselective embolization of all branches of the superior rectal artery. The transradial approach was most frequently performed compared to transfemoral access. After the procedure, no signs of ischemia were identified; three minor complications were observed. The mean follow-up was 18.5 ± 6.0 months. At the last follow-up, the mean increase of hemoglobin for patients was 1.2 ± 1.6 g/dL. Three patients needed transfusions during follow-up for recurrent hemorrhoidal bleeding. The Hemorrhoidal Disease Symptom Score and Short Health Scala decreased from 11.1 ± 4.2 to 4.7 ± 4.6 (*p* < 0.0001) and from 18.8 ± 4.8 to 10.2 ± 4.9 (*p* < 0.0001), respectively. Patients who had given up on their daily activities due to anemia have returned to their previous lifestyle. Emborrhoid seems to be a safe and effective option for the treatment of bleeding hemorrhoids in frail patients. The low complication rate and the significant reduction of post-defecation bleeding episodes are related to the improvement of the hemorrhoidal symptoms and patients’ quality of life.

## 1. Background

Rectal bleeding is one of the main chronic symptoms of hemorrhoidal disease that severely affects patients’ quality of life. It can cause severe anemia and drastically compromise the general state of health, especially in more frail patients. In order to reduce hemorrhoidal bleeding, phlebotonics and/or topical treatments (corticosteroids or anti-inflammatory agents) often represent the first therapeutic approaches, such us the office-based procedures (rubber band ligation, infrared photocoagulation, radiofrequency and sclerotherapy). These treatments are highly well tolerated, with a low complication rate, but often provide only short-term relief [1,2].

In all failures of the conservative management and severe anemia, surgery has to be considered. Several techniques have been proposed to treat symptomatic hemorrhoids, and the Doppler-guided hemorrhoidal artery ligation procedure (DG-HAL) is one of the most frequent minimally invasive approaches used [3,4]. The physiopathologic impact of DG-HAL on hemorrhoidal hyperflow is demonstrated by our recent study, which analyzed the hemodynamic effects after 12 months from the dearterialization. The study confirmed a significant decrease of the mean residual peak systolic velocity, pulsatility index and resistivity index with a drastic reduction of bleeding episodes [5].

Variable long-term results regarding the DG-HAL are reported in the literature, probably due to differences in the devices and stitches used, number and height of arterial ligations [6,7].

Based on the same principles, more recently, the Emborrhoid technique has proposed to perform a superselective endovascular occlusion of the terminal branches of the superior and middle rectal arteries. It is an innovative non-surgical technique for the treatment of severe hemorrhoidal bleeding [8].

Thus far, several studies have been published that include patients with heterogeneous baseline characteristics in terms of: hemorrhoidal disease degree, severity of bleeding and patient comorbidities. Clinical indication to perform Emborrhoid, patient selection, and its long-term impact have not been fully investigated [9].

The main objectives of our prospective study were to increase data regarding safety and efficacy of the Emborrhoid technique in frail patients with anemia and to assess the changes of hemorrhoidal disease symptoms and its impact on patients’ quality of life.

## 2. Materials and Methods

### 2.1. Study Design and Study Population

The study is reported according to the Strengthening the Reporting of Observational Studies in Epidemiology (STROBE) statement for cohort studies [10]. It was approved by the ethical committee of the Fondazione Policlinico Universitario Agostino Gemelli IRCCS, Rome, Italy; code: 3705.

From September 2020 to February 2022, all consecutive patients with bleeding hemorrhoids and anemia were evaluated to be enrolled in this prospective observational single-center study. All patients gave their written informed consent.

Demographic data, patient’s history, laboratory exams and previous blood transfusions were collected during the first visit. Therefore, a complete preoperative work-up (physical examination, endoscopy, and unenhanced and contrast-medium enhanced abdominal angio-CT) was performed in order to evaluate inclusion/exclusion criteria.

Therefore, all patients underwent a multidisciplinary pre-evaluation by the colorectal surgeons, the interventional radiologists, anesthesiologist and cardiologist to assess the risks and share the indication.

We included only patients older than 18 years of age and affected by symptomatic hemorrhoidal disease with anaemia and high surgical risk due to severe cardiovascular, respiratory and neurological disorders, cirrhosis/portal hypertension, antiplatelet and/or anticoagulant therapy and ASA ≥ 3. Patients ineligible for hemorrhoidal surgery with contraindication to general and/or spinal anaesthesia, or in the case of anal stenosis, were also considered for the protocol. If patients were on anticoagulants, they had to be able to stop medication temporarily prior to transarterial chemoembolization to obtain an INR < 1.5 at the time of the procedure.

On the other hand, we considered as exclusion criteria the following conditions:

Platelet count < 40,000/μL and/or international normalized ratio > 1.5, severe renal impairment, severe allergy or intolerance to any contrast media or chemotherapeutic drugs not controlled with medication, no available vascular access and/or absence of pulse in femoral or radial arteries, previous rectosigmoid resection, diagnosis of colorectal cancer, colonic angiodysplasia, inflammatory bowel disease, proctitis, or acute anorectal sepsis and family, psychological, social, or geographical circumstances preventing the patients from undergoing follow-up and from complying with protocol procedures.

### 2.2. Treatment

All treatments were performed by two experienced interventional radiologists (with more than 5 and 15 years of experience), using patient monitoring and anaesthesiologist assistance, in an angiographic suite with characteristics of an operating room. No antibiotic was administered before or after the procedure. Diagnostic angiography was performed under local anaesthesia (10 mL Lidocaine) using the Seldinger technique through the femoral or radial artery (based on operator preference for a single case) with a 5 Fr introducer sheath. Superselective catheterization of the superior rectal arteries was then performed using a coaxial technique, placing a 2.4 or 2.7 Fr Progreat microcatheter (Radifocus, Terumo, Rome, Italy). The coils used for the embolization were Interlock and IDC detachable embolization coils, to offer precision and control combined with thrombogenicity and flexibility, ranging between 4 and 7 mm in diameter, and 10–20 cm long (Boston Scientific, Marlborough, MA, USA). Also evaluated were all potential anastomoses with other feeder arteries (middle rectal or inferior rectal branches), advancing the 5 Fr Optitorque Multipurpose in the iliac internal artery and carrying out superselective injection into each branch using a coaxial technique with the same microcatheter. The anastomoses identified were embolized only in the cases in which they were among the main arteries feeding the hemorrhoidal plexus. All patients underwent haemostasis of the puncture site (radial or femoral). They were observed for 2 h (in the case of the transradial approach) or 6–8 h (in the case of the transfemoral approach) and discharged, if no complications arose, within 24 h (only in the case of the radial approach) or 48 h. Patients who had undergone the radial approach were managed in an outpatient setting, as reported in our previous pilot study [11].

Treatments needed to resolve any complications and recurrences were recorded.

Perioperative minor/major complications and death occurring within seven days from the treatment were evaluated and categorized according to the common terminology criteria for adverse event [12].

### 2.3. Endpoints and Definitions

The aim of our study is to report data on safety, efficacy and patients’ quality of life after the Emborrhoid technique in patients unsuitable for surgery with severe anemia due to hemorrhoidal bleeding. Technical success was defined as the correct and complete coil deployment, with a complete lack of opacification of the terminal branches of the superior/middle rectal arteries.

Major complications were defined as events determining substantial morbidity and disability, increased level of care, or lengthened hospital stay; all other complications were considered minor.

Follow-up was scheduled at 1, 6 and 12 months, and once a year thereafter, including a physical examination and a pulse check at the vascular access site.

Transfusions and serum hemoglobin level variations were registered at 6 and 12 months.

Patients who experienced new onset of anemia during the follow-up were regarded as a treatment failure.

The Hemorrhoidal Disease Symptom Score (HDSS) and Short Health Scale (SHS) [13] were completed before the procedure after 1 month, 6 months and 1 year during control visits or via telephone interviews. The HDSS is based on five different parameters characterizing the hemorrhoidal disease, with a grading from 0 (no symptoms) to 4 (daily presence) for each symptom. The total score of all five parameters was used to evaluate the patient’s condition: 0 indicated the total absence of a symptom, while a score of 20 represented the worst clinical conditions. The SHS is a QoL-based score that includes information on symptoms severity, impact on daily activities, patients’ concerns, and personal feeling of well-being ranging from 1 (optimal clinical conditions) to 28 (worst clinical scenario). They have been validated only in the English language and have been internally translated for clinical practice. As a mitigation, surgeons were present to guarantee adequate comprehension of the questions.

### 2.4. Statistical Analysis

Continuous variables were reported as mean and standard deviation. Categorical variables were described by count and percentage. Comparisons between groups were performed using the Wilcoxon test for continuous variables and the Fisher exact test for categorical data. Statistical significance was defined for *p* < 0.05.

Statistical analysis was performed with SPSS statistics for Windows version 23.0 software (IBM, Armonk, NY, USA).

## 3. Results

From September 2020 to February 2022, 24 patients were evaluated for embolization. After complete pre-treatment work-up, three patients were excluded from our protocol, in particular, two patients for severe heart failure and one patient due to right colon cancer.

### 3.1. Patients Characteristics

After multidisciplinary discussion, 21 patients with high surgical risk (16 male, 76.2%; mean age 72.2 ± 10.9 range 47–92 years) were candidates for Emborrhoid. As highlighted in Table 1, 18 patients were affected also by severe cardiovascular disease, six patients presented with obstructive pulmonary disease, two patients had hematologic or immunological disorders, five patients had mild chronic kidney failure, obesity plus metabolic syndrome in 17 patients, and one case of paraplegia. Nine patients were in anticoagulant therapy, six in antiplatelet therapy and one in anticoagulant plus antiplatelet therapy. Six patients (28.6%) had previously undergone a hemorrhoidectomy and presented with recurrence of bleeding. According to Goligher’s classification, II degree was identified in eight patients, III degree in 10 and IV degree in three cases. Flavonoids and iron were already used by 95.2% of patients. Nineteen (90.5%) patients were referred for daily post-defecation bleeding. Pre-treatment blood transfusions were required for 52.4% of patients with severe anemia.

### 3.2. Operative and Perioperative Data

Superselective embolization of all branches of the superior rectal artery was performed in all patients, obtaining a technical success rate of 100%. Left radial access was successfully performed in 17 cases, and a switch from radial access to femoral access was reported in one case (cross rate 4.8%). Total examination times ranged from 40 to 50 min with low radiation doses and contrast volumes administered between 70 and 80 mL.

Distal embolization of the middle rectal branches was also required in three patients (bilateral in two patients and unilateral in one), using the same coils (Figure 1).

Haemostasis was performed using a TR band radial compression device (Terumo medical corporation) in 13 patients who underwent transradial access, a 6 Fr Angio-seal vascular closure device (Terumo medical corporation) in four patients and manual compression of the femoral artery in four patients.

Stability of vital signs was registered in all patients, with adequate respiratory function. All patients had an adequate level of orientation and alertness, ability to tolerate a clear liquid diet, and absence of significant pain, with mild pain in one patient only, self-limited without requiring medical therapy.

Bowel movements were registered on the first and second days after the procedure, without severe bleeding.

All patients were discharged without complications, four patients after 48 h (all cases of femoral approach) and 17 patients who underwent transradial Emborrhoid, were managed in an outpatient setting, and discharged within 24 h of the procedure.

No signs of ischemia, radial or femoral pulse absence occurred. Three minor postoperative complications were observed (one ecchymosis, one arm pain, one pseudoaneurysm of radial artery), safely managed with conservative treatment. The overall complication rate was 14.3%.

Transient ischemic attacks, reversible ischemic neurologic deficits, and stroke, defined as a new, persistent neurologic disability lasting >24 h, were never registered.

All patients had reduction of hemorrhoidal congestion, no anal pain occurred after the procedure, tenesmus was referred by one patient.

No variations of anorectal sensitivity and of faecal continence were reported. Anal ulceration, ischemia and anorectal perforation were never diagnosed after the procedure and at the follow-up visits.

### 3.3. Middle- and Long-Term Follow-Up

Mean follow-up was 18.5 ± 6.0 months. One-year follow-up was assessed in 17 patients (81%).

Patients who daily experienced post-defecation bleeding episodes decreased from 19 to four. At the last follow-up, the mean increase of hemoglobin for a patient was 1.2 ± 1.6 g/dL. Three patients who experienced new onset of anemia during the follow-up were regarded as a treatment failure (overall recurrence rate was 14.3%). Two patients needed transfusions within one month and one patient within six months from the embolization. Consequently, one patient underwent surgical dearterialization, and two patients received sclerotherapy.

A case of death, for other causes (leukemia), was registered after five months from embolization.

At the 12-month follow-up visit, HDSS and SHS decreased from 11.1 ± 4.2 to 4.7 ± 4.6 (*p* < 0.0001) and from 18.8 ± 4.8 to 10.2 ± 4.9 (*p* < 0.0001), respectively, with a significant improvement of hemorrhoidal symptoms, except for itching. All patients referred the decrease of hemorrhoidal symptoms, especially for the bleeding severity (*p* < 0.0001), soiling (*p* = 0.002), frequency of postdefecatory hemorrhoidal prolapse (*p* = 0.006) and anal pain (*p* = 0.001); whereas, no significant difference was reported for the perianal itching (*p* = 0.111). The state of severity and anxiety/worry related to the hemorrhoidal disease referred by patients were greatly reduced (*p* < 0.0001 and *p* = 0.002) and associated with an improvement of general well-being (*p* = 0.006) (Figure 2).

Patients who had given up their daily activities due to hemorrhoidal disease have returned to their previous lifestyle. Moreover, the low complication rate and the significant reduction of post-defecation bleeding episodes are related to the improvement of the hemorrhoidal symptoms and patients’ quality of life.

## 4. Discussion

Over the years, the therapeutic range available to treat the patient affected by hemorrhoidal disease has expanded, allowing a more individualized treatment [14]. Emborrhoid is a relatively new nonoperative treatment of hemorrhoidal bleeding, based on the same physiopathological principles of hemorrhoidal dearterialization. It consists in embolization with coils or microparticle of terminal branches of the superior and middle rectal arteries via the endovascular route. This approach was proposed by Vidal as a valid option for the treatment of chronic hemorrhoidal bleeding in patients with contraindications for surgery [8,15].

Although a recent case report described an event of sigmoid ischemia/stenosis after microparticles SRA embolization [16], several papers seem to demonstrate that this endovascular approach is safe and effective. Many authors supported the Emborrhoid technique either in the case of contraindication for surgery or as first line treatment in low grade of hemorrhoidal disease [17,18,19,20]. In these studies, the Emborrhoid technique has traditionally been performed on an inpatient basis, generally using the right femoral route, but the heterogeneity in patient selection (also in the same study) and pre- or postoperative assessment makes the results difficult to compare. Two reviews of literature underline the lack of homogeneous and comparable data, in particular, the evidence regarding appropriated indication, middle/long-term results, and cost-effectiveness remains insufficient [9,21].

Our prospective study aims to increase middle/long-term clinical results after embolization in selected patients with hemorrhoidal bleeding and to assess the impact of embolization on hemorrhoidal symptoms and patients’ quality of life. Only patients with anemia and high surgical risk due to severe cardiovascular, pulmonary or absolute contraindication to anesthesia were enrolled. The technical success rate was 100%. Seventeen patients were treated via transradial access and four via transfemoral (including a case of a switch from radial to femoral access) without major intra- and perioperative complications. The overall recurrence rate was 14.3%.

Other studies investigating the impact of embolization in frail patients are available in the literature. In details, Moussa et al. reported a success rate of 72% in patients with contraindication for surgery (30 consecutive patients with hemostatic disorders, previous hemorrhoidectomy and affected by inflammatory bowel disease), after a single procedure or two procedures, with a significant improvement of the French bleeding score, of the general symptom score and of the patient’s quality-of-life score, without complications (only a notable event of diarrhea self-limited) [22]. Similarly, in a case report of Venturini et al., two patients with heart disease and contraindication to suspension antiplatelet or anticoagulation therapy underwent coil embolization of SRA. No further blood transfusions and no complications were recorded at the one-month follow-up [23]. Two other case studies were published regarding the hemorrhoidal embolization in patients with portal hypertension. Both studies confirmed the safety and efficacy of the endovascular procedure in the short term, without major complications [24,25]. All these papers show encouraging perioperative results but lack long-term follow-up.

In our study, the mean follow-up was 18.5 ± 6 months. One-year follow-up was assessed in 17 patients (81%). Neither anorectal ischemia, ulceration nor perforation was recorded in short-term and one-year follow-ups. Three patients had new-onset anemia and received blood transfusions; of these, one patient underwent DG-HAL and in two cases sclerotherapy was performed. Considering the entire follow-up period, we recorded the relapse within the first six months.

Analyzing the HDSS and SHS scores showed a significant improvement of hemorrhoidal symptoms (expect for itching) and of the patient’s quality of life. These positive results were obtained performing a standardized approach, using dedicated devices and based on an adequate screening process. Notably, the majority of our patients were treated in an outpatient setting and discharged within 24 h. No need for pre-procedural groin preparation, less post-procedural discomfort at the access route, and reduced limitations for the patient in performing basic activities are the main drivers for this relatively new approach. Therefore, in our opinion, any refinement in embolotherapy that reduces the impact on a patient’s life, such as performing embolization during a single outpatient session, may bring benefits in terms of quality of cure and cost.

The limitations of our study are represented by the single-center study and by the small number of patients.

## 5. Conclusions

The Emborrhoid technique seems to have a promising role in the clinical practice for the treatment of patients with chronic hemorrhoidal bleeding and high surgical risk. Future multicenter prospective trials performed on larger populations will be necessary to more thoroughly evaluate the clinical efficacy of Emborrhoid techniques and to clarify the indication and clinical management.

## Figures and Tables

**Figure 1 jcm-11-05533-f001:**
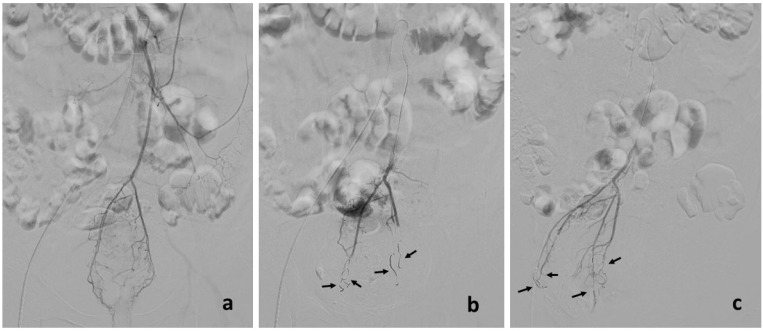
Diagnostic phase with hemorrhoidal arteries evaluation (**a**). Four Interlock and IDC coils (arrows) were positioned in the distal branches of the superior hemorrhoidal arteries ((**b**) = anteroposterior radiographic view; (**c**) = latero-lateral radiographic view).

**Figure 2 jcm-11-05533-f002:**
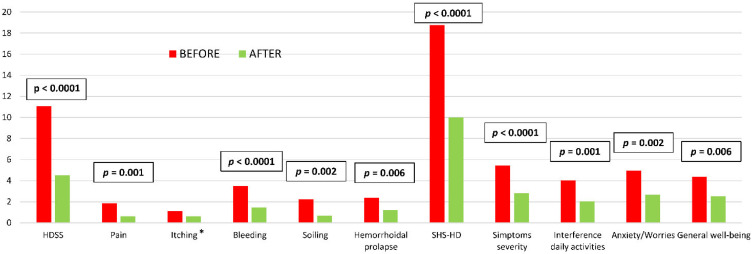
Comparison between HDSS and SHS-HD scores collected before Emborrhoid technique and at last follow-up visit (Wilcoxon test). * *p* = 0.111. HDSS = Hemorrhoidal Disease Symptoms Score; SHS-HD = Short Health Scale for Hemorrhoidal Disease.

**Table 1 jcm-11-05533-t001:** Baseline patients’ characteristics.

	** *n* ** **(%)**
Patients	21
Ratio M:F	16:5
Age (years) *	72.2 (10.9)
Comorbidities	
Cardiovascular	18 (85.7)
Respiratory	6 (28.6)
Hematological/immunological	2 (9.5)
Mild chronic kidney failure	5 (23.8)
Obesity/dyslipidemia	17 (81.0)
Paraplegia	1 (4.8)
History of smoking/smoker	10 (47.6)
Antiplatelet therapy	6 (28.6)
Anticoagulant therapy	9 (42.9)
Antiplatelet + anticoagulant therapy	1 (4.8)
Previous hemorrhoidal surgery	6 (28.6)
Goligher’s classification	
Grade II	8 (38.1)
Grade III	10 (47.6)
Grade IV	3 (14.3)
Flavonoids before embolization	17 (81.0)
Iron therapy before embolization	10 (47.6)
Blood transfusion before embolization	11 (52.4)

* Data are shown as mean ± standard deviation. M = male; F = female.

## Data Availability

The data presented in this study are available on request from the corresponding author.

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
