# Peer review of "The Emborrhoid Technique for Treatment of Bleeding Hemorrhoids in Patients with High Surgical Risk†"

_jcm, 2022, doi:10.3390/jcm11195533_

Round 1

Reviewer 1 Report

Thank you for this manuscript regarding the Emborrhoid technique. I appreciate your paper and the design. The study design would be strengthened with more patients included and longer follow up, a second manuscript with updated information is warranted given the small number of enrolled patients and few that made it to 12 months of follow up. 

Author Response

Thanks for your comments.

We edited the english with the support of Mrs Kelly Baron (UK).

We updated the follow-up of our case series.

Reviewer 2 Report

Interesting and well planned clinical study.

Well writting and methodological ly correct

Only two observations: in the introduction, the paragraph on DG-HAL should be reduced, since it only raised phisiopathological aspects. Other minimally invasive techniques such as radiofrequency are no mentioned in theses cases.

As it is a prospective study of safe and security there is a lack of endoscopic review of the patients to assess mucosal changes in the distal rectum and anal canal.

Author Response

Thanks for your comments.

We included the radiofruency in the introduction and reduced the THD description.

We reported no change in anorectal mucosa at follow-up visit and anoscopy, moreover we updated the follow-up in August.

Reviewer 3 Report

Thanks for this team to share their results concerning this salvage option for haemorrhoidal disease in very frail patients.

Just one point is missing for me : as you have different f-up evaluation dates, could you give information about the evolution of the results according to time elapsed since intervention in individual patients. Or, do you see a difference in efficacy according to the length of f-up amongst the entire population.

Congratulations for this innovative work.

Author Response

Thanks for your comments,

we edited the language with the support of Kelly Baron (UK).

we updated the follow-up of our case series.

mean follow-up  was 18.5±6.0 months,  and one year follow-up was assessed in 17 patients (81% of patients). The value  of Hb remained stable. No transfusions were reported. All the relapses were recorded within 6 months from emborrhoid.